# Two-Dimensional Path Planning Platform for Autonomous Walk behind Hand Tractor

Padma Nyoman Crisnapati 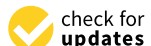 and Dechrit Maneetham *

Department of Mechatronics Engineering, Rajamangala University of Technology Thanyaburi,
Pathum Thani 12110, Thailand
* Correspondence: dechrit_m@rmutt.ac.th

**Abstract:** The use of autonomous vehicles in agriculture has increased in recent years. To fully automate agricultural missions, particularly the tillage process using the walk-behind hand tractor, the path planning problem for the robot must be solved so that all points in the intended region of interest may be traced. The current planning algorithm has been successful in determining the best tillage path. On the other hand, the algorithm ignores the path built using the dynamic starting point, finish point and path distance. We propose a path planning concept for back-and-forth path patterns. Our algorithm employs a novel approach based on Laravel and Google Maps, which considers the user's desired distance interval, start point, and finish point. We demonstrated auto-generating vertex-edge pathways in this research. Field trials using a walk-behind hand tractor in a plowing mission have been successfully conducted to validate the accuracy of the resulting waypoint coordinates.

**Keywords:** path planning; back-forth path; hand tractor; tillage; unmanned ground vehicle

## 1. Introduction

Mobile robots have been effectively used to perform vital unmanned operations in various situations during the last few decades, including military, industrial, security, and agricultural operations [1]. Mobile robots are increasingly being used in modern agriculture because the number of farmers is decreasing, necessitating more effective farming practices through agricultural mechanization. Agriculture has been mechanized in most field operations, including tillage, transplantation, agrochemical application, harvesting, and drying. While most agriculture operations are being explored for automation, Indonesian farmers' tillage methods are not yet automated.

Path Planning [2] is a critical problem that must be solved before a mobile robot may navigate and explore autonomously; this kind of robot is usually known as an Unmanned Ground Vehicle (UGV). The UGV can search for pathways based on their start and finish points, the surrounding environment, and specified parameters. With path planning, the UGV can save time and significantly reduce the UGV's wear and tear and the associated costs. The critical significance of path planning for UGVs is an intriguing subject of study.

UGV applications have exploded in popularity during the last five years. Survey missions for two-dimensional (2D) coverage have demonstrated exceptional performance among diverse uses. For instance, two-dimensional and three-dimensional mapping [3], search and rescue [4] in combination with an Unmanned Aerial Vehicle for disaster and emergency management [5], or precision agriculture [6]. The survey mission performed by UGVs in agriculture is generally separated into two stages: (i) the preparation stage, during which the vehicle is selected, the embedded system configured, and the path is planned; and (ii) the execution stage, during which the vehicle operates autonomously and gathers data. Path planning must be completed to fully automate the operation, which is described as calculating paths for the robot to traverse the Region of Interest (ROI) [7]. The problem's complexity has been determined as NP-Hard [8]. However, assuming that traveling in

a straight line is the quickest and most efficient way to traverse the entire landscape, a sophisticated solution involves reducing the problem and computing the path with the least tracing, e.g., [9,10]. While several of these studies have produced significant results, they do not account for the path factors associated with the user's start and finish points. The puddler widths can be changed because the UGV is configured as a walk-behind hand tractor. The path planning must account for additional criteria such as interval tillage line distance from the user. Taking these three factors into account will affect the accuracy of land management while utilizing various models of tractors; this requires a path design that considers the mission's distance interval and the mission's starting and ending sites.

Due to limited field resources, efficient path computation is required; this can be accomplished by moving the tractor in a straight line, forming a back-and-forth path (BFP) [11]. This study analyses convex polygon boustrophedon routes, a fast algorithm for determining BFP coverage on an ROI, considering the distance interval, start, and finish points (Figure 1). The decomposition of boustrophedon cells is a widely used technique for coverage. The cells of the boustrophedon are filled in a simple backward and forward motion. Once each cell is closed, the entire environment is sealed [12]. As a result, the scope is narrowed to identify the whole path through the graph, reflecting the boustrophedon decomposition's cell proximity connection. This strategy is well-suited for plowing missions with defined start and finish points and coverage.

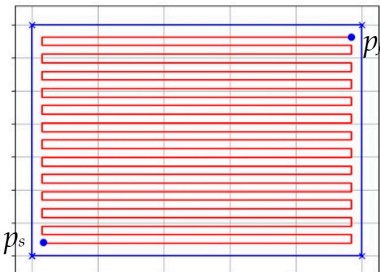

**Figure 1.** Proposed concept for path planning, the red line represents the generated path based on the starting point (*ps*), finish point (*pf*), and the ROI (blue polygon).

Over the last few decades, path planning problems have been addressed extensively; nevertheless, the available literature focuses primarily on UGV and UAV difficulties. Various strategies and algorithms can be employed with ground robots; generally, path planning based on ROI can be classified as solid or border representations. NP-Hard problems are used as a base to path planning with solid-representation approaches; emphasis is on interiors rather than regions [8]. Generally, the interior of an environment is represented as a grid of binary values or probabilities [13]. This method is also known as the grid-based method [7]. This resolution-complete approach takes a significant amount of computer power for high-resolution maps. The grid-based process makes use of a neural network in which the region is the input [14], divides it into triangular cells, and in conjunction with a region growth algorithm, generates a path [3], utilizing a Turing machine [15] or by using a genetic algorithm [16]. In addition to solid representation, the boundary representation approach defines the region of interest (ROI) depending on the geometry of the polygon. The polygonal model is adopted in this study. Numerous literature evaluations exist for alternative path-planning approaches [7].

One of the most extensively used polygonal representations is boustrophedon cellular decomposition (BCD); this method provides a path-planning solution for traversing the polynomial world [12]. BCD generates a line using the idea of cells and then explores each cell using this manner, except that the robot's direction of movement is always the same. Huang [9] developed another coverage approach for mine-operating robots that uses alternate orientations perpendicular to the ROI range instead of polygons. However, this method ignores the trip distance between the starting point and the ROI. Numerous studies have been conducted on the covering of non-convex terrain. This strategy's fundamental

difficulty is acquiring a near-optimal ROI partition and mapping coverage path to visit each division sequentially [17]; in some circumstances, other restrictions such as energy consumption are incorporated into the optimization process [18]. In this paper, we focus exclusively on the convex region.

Path planning with UGVs has been generally approached from the standpoint of land-based mobile robotics, but with novel functionalities. Most approaches have adhered to Huang's optimality criteria [9], where the ideal path is the one with the fewest paths. The techniques for calculating the convex polygons area are provided by [19]. The convex covering area method is presented in research [10]. Numerous researchers have also used path planning in agriculture, including UGV, to determine the ground features of greenhouses [20] by utilizing the BFP technique and a differential robot.

Further, [21] created a robot for precision pollination in a greenhouse and implemented it on a differential robot. Path planning for an unmanned ground vehicle in conjunction with aerial images utilizing the A* search method in graphs with gradient optimization of the descent to smooth the trajectory [22]. Large tractor machines are used; this work optimizes the harvest area of a combined harvester robot for wheat or rice using convex and concave polygon fields [23].

In short, the prior approach ignored the dynamic size of the tractor puddler, limiting its application to a single vehicle, and the approach was limited to establishing the beginning point and the tractor with huge dimensions. As a result, the initial approximation is inadequate. A more accurate path can be obtained by combining the dynamic starting inputs for the interval distance, start point, and finish point and then applying these to the walk-behind tractor. This paper is a continuation of our previous work, TROLLS: Tractor Controlling System for Walk-Behind Hand Tractors, presented in its first version [24,25]. The initial version can control the tractor remotely without automation. This article presents novel results in an open-source web-based platform and Google Maps for planning rectangular polygon paths that consider the width of the puddler in addition to the mission distance interval's start point and end point as criteria. This path planning is the first step towards an autonomously operating tractor engine validated in field trials using the tractor G1000 manufactured by Quick. The plowing mission was carried out as a validation of the success of the path-planning platform. The tractor used in the field test is equipped with an embedded system platform that allows the tractor to move autonomously based on the path planning platform's waypoints.

## 2. Materials and Methods

### 2.1. General Definition and Notation

The tractor in this study moves in a two-dimensional planar plane, where *L* is a linear combination (Figure 1), as in the Formula (1). The distance between point *a* and line *L* is the perpendicular distance between *c* and *a* point on *L* (2). The distance between parallel lines $L_1$ and $L_2$ is denoted by (3). Equation (4) shows a line segment as a horizontal line linking two points, *c* and *d* [26]. Abbreviations and symbols used in the article are listed in Table 1.

$$[c, d] = L(\alpha) = (1 - \alpha)\, c + d \rightarrow \alpha \in R \tag{1}$$

$$dst\,(a, L) = dst\,(a, b) \rightarrow b \in L, [c, d] \perp L \tag{2}$$

$$dst\,(c \in L_1, L_2) \tag{3}$$

$$\overline{cd} = T\,(\alpha) = (1 - \alpha)\, c + \alpha d \rightarrow 0 \leq \alpha \leq 1 \tag{4}$$

### 2.2. Region of Interest (ROI)

As seen in Figure 1, The ROI in this study is in the form of a 2D planar (convex polygon) (5), where a collection of vertices and edges is represented by the Formulas (6) and (7) [11,26]. *L* is the support line formed by the intersection with the polygon boundary

line and forms a pair of antipodal points. The distance between the two support lines is created on the same polygon and is named width [19].

$$Q = \{V, E\} \tag{5}$$

$$V = \{1, \ldots, n\} \tag{6}$$

$$E = \{(1, 2), \ldots, (n-1, n), (n, 1)\} \tag{7}$$

**Table 1.** Abbreviations and symbols are used in this article.

| Symbol | Description | Symbol | Description |
|--------|-------------|--------|-------------|
| $R^2$ | Two-dimensional (2D) planar tractor work area | $\varphi_1$ | Latitude of the initial point |
| $(x, y)$ | Within the work area, a point is a location | $\varphi_2$ | Latitude of the final point |
| $L$ | Line as a linear combination of two points | $\Delta\varphi$ | $\varphi_2 - \varphi_1$ |
| $a, b, c, d$ | Four points within a Region of Interest (ROI) | $\lambda$ | Longitude |
| $Q$ | The Region of Interest | $\lambda_1$ | Longitude of Initial Point |
| $V$ | A collection of points on a plane (vertices) | $\lambda_2$ | Longitude of Final Point |
| $E$ | A group of edges | $\Delta\lambda$ | $\lambda_2 - \lambda_1$ |
| $A(Q)$ | Polygon area | *earth's_radius* | 6371 km |
| $Ix$ | Tillage footprint length | $d$ | Distance between two points |
| $Iy$ | Tillage footprint width | $ps$ | Starting point |
| $p$ | Tractor position | $pf$ | Finish point |
| $p1, p2$ | Two points on the coverage area of the puddler | $p0, \ldots, pn$ | Knot edge path |
| $C(T)$ | The coverage area of a tillage line | $dx$ | Distance between tillage line |
| $\varphi$ | Latitude | $\delta$ | Declination angle |
| $dst$ | Distance between two parallel lines | | |

### 2.3. Unmanned Ground Vehicle

When the tractor is used for tillage operations, it is equipped with a puddler that points directly to the ground at a consistent harrow height. The tractor location is defined as $p = (x, y)$ in $R^2$, which allows for the definition of the tractor path as $s(t): R \rightarrow R^2$.

### 2.4. Two-Dimensional Leveller/Puddler Footprint

When the tractor plows an area, the tractor departs from point ps, following the path by carrying a puddler and heading to the finish point *pf* (see Figure 1). During the tillage process, the puddler forms an area known as the tillage footprint of $Ix \times Iy$ so that the tillage footprint can be represented by Equation (8). If s is the tillage path, then Equation (9) is the coverage area ($\rho$) of the puddler, and each waypoint of $Q$ is represented in Equation (10). The tractor tillage mission overview is presented in Figure 2.

$$Tl(p) = Ix \times Iy \tag{8}$$

$$C(s) = \cup p \in s\, Tl(p) \tag{9}$$

$$A(Q) \subseteq C(\rho) \tag{10}$$

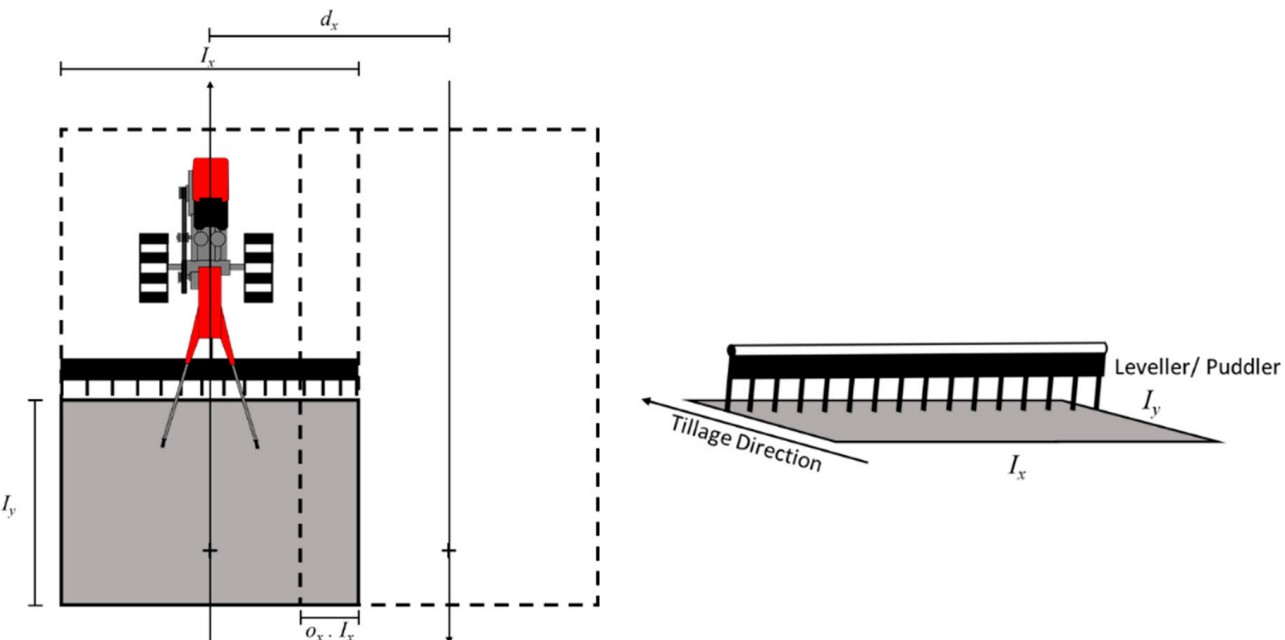

**Figure 2.** Walk-behind hand tractor tillage mission from a perspective view. The top perspective is depicted on the left. Lines with arrows indicate the direction of tillage. The gray box denotes the area crossed by the puddler, referred to as the *Tl(p)* tillage footprint. After completing one tillage line, the tractor will proceed to the next. *dx* denotes the distance between tillage lines. The right image depicts a view from a lateral position.

*2.5. Back and Forth Path (BFP)*

In path planning, many possible path forms are created [8]. Therefore, particular patterns such as spirals, zigzags, stars, or back and forth are the solution to these problems. In this study, the BFP is implemented on a tractor that moves in a straight line at the ROI. The advantages of BFP allow the tractor to keep the puddler stable and make it easy for autonomous vehicles to follow the tillage line.

Tillage line *T* is a linear combination Equation (11), which means that $\alpha \in R$ and $0 \leq \alpha \leq 1$ as seen in Figure 3. $C(T)$ is the region enclosed by $T_{left}$ and $T_{right}$. These two lines are parallel to *T* by an $\frac{l_x}{2}$ distance. When tillage occurs at an ROI, one tillage line is insufficient to cover the entire area, so plowing with a BFP pattern is used in Equation (12). Equation (13) is used to calculate the coverage area of *Q*.

$$T(\alpha) = (1 - \alpha)\, p_1 + \alpha\, p_2 \tag{11}$$

$$P = \{T_1, \dots, T_n\} \tag{12}$$

$$A(Q) \subseteq \cup F \in P\; C(T) \tag{13}$$

These calculations deduce that BFP is a collection of sequential points used as a plowing route boundary or waypoint, which we connect with a straight line between tillage lines. This research aims to develop a path planning platform for rectangular ROI convex polygons and validate the autonomous walk-behind hand tractor's algorithm.

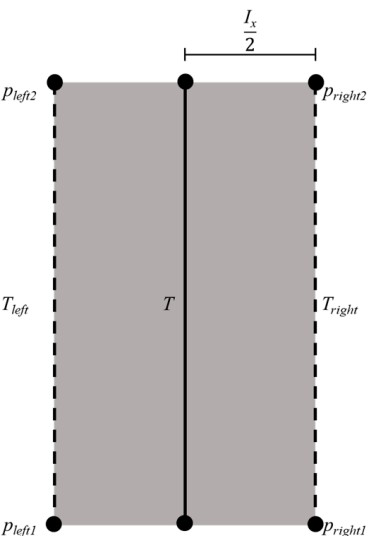

**Figure 3.** Area covered when the tractor crosses the T tillage line.

## 3. Embedded System Platform

Walk-behind tractors (affectionately referred to as walking tractors, pedestrian-controlled tractors, or power tillers) are prevalent in small and medium-sized rural communities. This type of tractor offers several advantages, including its low price and ease of use. Additionally, these tractors can perform various agricultural tasks, including harvesting, crop protection, irrigation, threshing, and transporting [27]. A G1000 Quick tractor equipped with a boxed embedded control system was used in this study. The tractor's wheels turn due to a clutch handle being pulled, as illustrated in Figure 4. This platform is used to validate the path planning platform.

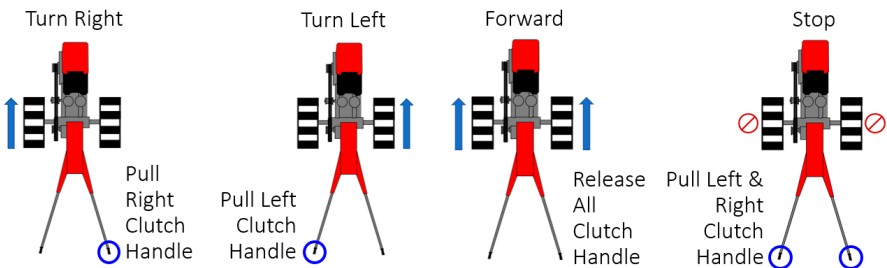

**Figure 4.** G1000-Quick Walk-Behind Tractor movement control.

An autonomous navigation system enables tractors to navigate between waypoints autonomously [28]. We are attempting to construct an additional validation platform. First, the list of waypoints generated by the platform informs of the final Global Positioning System (GPS) coordinates (longitude and latitude). The platform will then build a set of reference pathways (straight lines) that connect each waypoint. Additionally, these coordinates are used in a control system to direct the robot's movement between waypoints along the reference route. The results obtained from the GPS and IMU sensors contain significant noise. This will interfere with the tractor's movement system; hence, the application of a filter in this system is essential. As an early observation, the Kalman Filter and the Butterworth Filter were compared. The results of applying these two techniques to both sensors are depicted in Figure 5. It is evident from the image that the application of the filter effectively reduces the noise and error that arises. Kalman filter was selected because of its superior input stability for the controller. The controller, not the tractor, determines the distance and heading. Figure 6 depicts the relationship between GPS sensors, IMU (Magnetometer), and Waypoint Techniques as inputs and outputs rather than controllers such as servo motor drives.

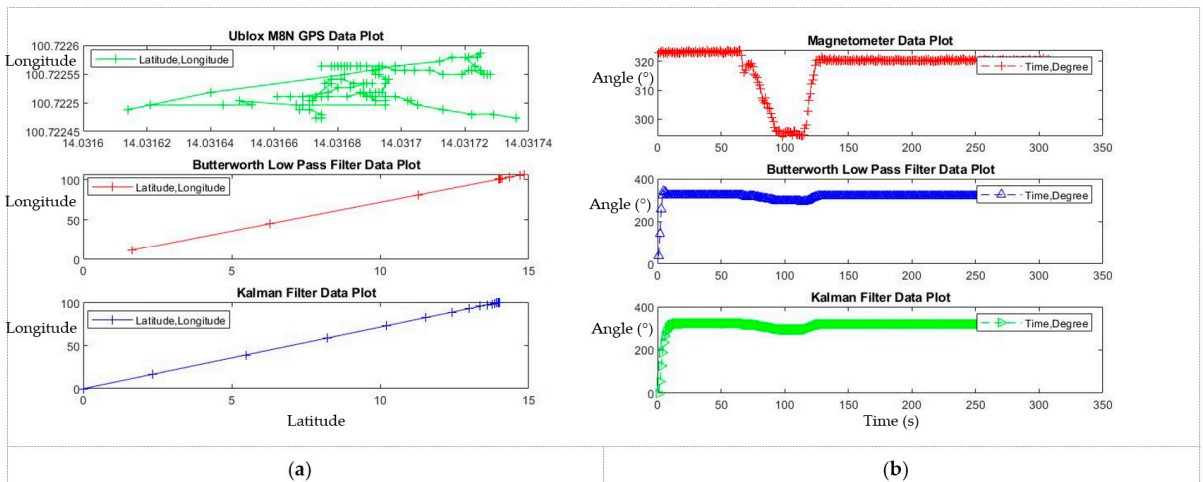

**Figure 5.** Comparison of noise and error correction data plot using filter. (**a**) GPS Data Plot; (**b**) Magnetometer Data Plot.

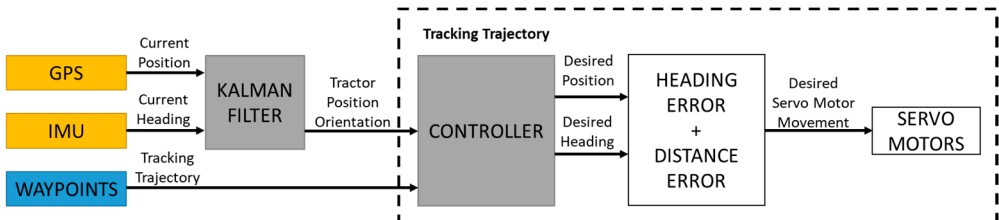

**Figure 6.** Waypoint system overview.

GPS technology is used to determine the current location of the tractor using a GPS receiver [29]. GPS technology is the backbone of the autonomous vehicle because it receives GPS coordinates from the location; the U-Blox Neo-M8N type sensor is used in this study. As indicated in Table 2, this type of GPS was chosen because it is economical for Indonesian farmers and offers a reasonable degree of precision.

**Table 2.** Neo M8N first fix and horizontal accuracy.

| Parameter | Condition | GPS | GLONASS | BeiDou |
|---|---|---|---|---|
| First fix time | Cold start | 29 s | 26 s | 27 s |
| | Hot start | 1 s | 1 s | 1 s |
| Horizontal accuracy | Autonomous | 2.5 m | 2.5 m | 2.5 m |
| | SBAS | 2.0 m | 2.0 m | 2.0 m |

After the tractor location is obtained, the distance between the current tractor position and the target position is calculated using the Haversine Equations (14)–(16) [30]. In addition to distance, bearing angle is an essential metric for autonomous robotics as it indicates the robot's direction and helps the tractor follow the correct path [30]. The angle of the tractor bearing concerning the initial latitude and longitude coordinates can be calculated using the Equations (17)–(19).

$$x = sin^2(\Delta\varphi/2) + cos \; \varphi1 \cdot cos \; \varphi2 \cdot sin^2 \; (\Delta\lambda/2) \tag{14}$$

$$y = 2 \cdot atan2(\sqrt{a}, \sqrt{(1-a)}) \tag{15}$$

$$d = y. \; earth's\_radius \tag{16}$$

$$x = sin(\Delta\lambda) \; cos(\varphi1) \tag{17}$$

$$y = cos(\varphi1) \cdot sin(\Delta\varphi2) - sin(\varphi1) \cdot cos(\varphi2) \, cos(\Delta\lambda) \tag{18}$$

$$Bearing\ Angle = atan2\ (x, y) \cdot 180/\pi \tag{19}$$

The heading angle is obtained from the magnetometer sensor found on the IMU. This sensor can provide an absolute correct direction, but its efficiency is reduced when a magnetic field other than the Earth's magnetic field is present [31]. One source of significant magnetometer error is the annual variation in the tilt of the Earth's axis of rotation and the Earth's rotation around the sun, also known as the declination angle. The declination angle can be determined using the Equation (20), where a day is the number of days remaining before January. The declination angle should be added to the sensor-derived compass direction [31]. This inaccuracy varies according to the sensor's location and can be found at https://www.magnetic-declination.com/ (1 February 2021). For instance, in Gianyar, Bali- Indonesia, the declination is 0°43′ EAST (POSITIVE). Autonomous UGVs require knowledge about their travel direction. The heading angle in the horizontal plane is defined as the clockwise angle from true north [30]. As a result, headings that rotate between 0° and 360° refer to true north (21).

$$D = sin - 1(sin(23{,}45°) \cdot sin\ (360/365 \cdot (day\text{-}81)) \tag{20}$$

$$heading = atan\ (Yh/Xh) \tag{21}$$

## 4. Results and Discussion

This research aims to compute the coverage path Equation (22) to obtain an edge-vertex path with a boustrophedon cellular decomposition pattern and test it on the Quick G1000 walk-behind tractor. Briefly, our goal is to determine the edge-vertex routes within the polygon, the start-finish points, and the distance between tillage lines.

$$\tau = \{p_s, p_0, \ldots, p_n, p_f\} \tag{22}$$

Figure 7 shows a path that forms an alternate direction when the tillage line generated is parallel to one side of the ROI. An edge-vertex path (EVP) is formed inside polygon $Q$. The EVP is formed by the intersection of waypoints and the $L_{tillage}$ line with polygon $Q$. In Algorithm 1, the inputs are the polygon $Q$, the initial vertex $b$, the antipodal vertex $c$ and $d$, and the distance between the tillage line $d_x$. $d_x$ is the first user input determined by the size of the puddler rather than the type. At the beginning process, the $L_{tillage}$ line is parallel to the sides $(a, b)$ that have been displaced perpendicularly toward the $c$ and $d$ directions. After intersecting with polygon $Q$, $L_{tillage}$ subtracts or adds $d_x$ to create $p_1$ and $p_2$. The next step is to combine the points into the path and connect them to the CalculateConnect function, which provides a perpendicular boundary between the two points. If the footprint intersects the polygon line, this method will be repeated and shifted to point $c$ or $d$. This algorithm returns the path $\rho = \{p_0, \ldots, p_n\}$, at the final step. EVP always begins from vertex $a$ and sweeps towards vertex $b$, $c$, and $d$.

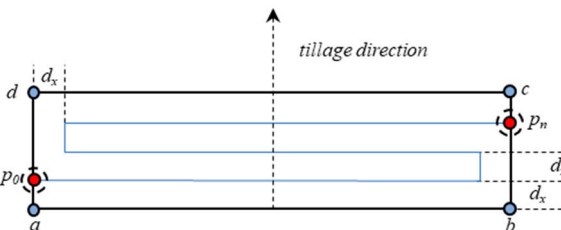

**Figure 7.** A diagrammatic representation of the suggested edge-vertex path algorithm for walking behind a hand tractor. The algorithm regularly adds points by recognizing the lines' intersection with a predefined region of interest (ROI).

**Algorithm 1:** Polygon ROI edge-vertex identification (GetPath($a$, $b$, $c$, $d$)). Calculating the BFP path begins with ($\rho = \{p_0, \ldots, p_n\}$) for the polygon ($Q = V$, $E$). The tractor begins its movement at vertex $a$ and sweeps in the direction of $c$ and $d$.

**Data:** $Q$, $dx$, $a$, $b$, $c$, $d$

**Result:** $\rho$

1.    $L_{tillage} \leftarrow$ CreateLine($a$, $b$);
2.    $L_{tillage} \leftarrow$ OffSet($L_{tillage}$, $d_x$);
3.    $\rho \leftarrow \varnothing$;
4.    **while** *Intersects*($C(L_{tillage})$; $Q$) **do**
5.        **if** (IntersectEdgesLeft) **then**
6.            $ip_1$; $ip_2$ IntersectEdges($L_{tillage}$; $E$) + $d_x$;
7.        **else if** (IntersectEdgesRight) **then**
8.            $ip_1$; $ip_2$ IntersectEdges($L_{tillage}$; $E$) − $d_x$;
9.        **end if**
10.       $\rho \leftarrow$ CalculateConnect($\rho$; $ip_1$; $ip_2$);
11.       $L_{tillage} \leftarrow$ OffSet($L_{tillage}$, $dx$);
12.   **end while**
13.   return $\rho$;

The result of implementing Algorithm 1 is a Laravel-based website platform that uses Google Maps for satellite map sources. The user can add a region of interest to the accessible map and specify the interval distance in meters, and the starting and ending points. After determining all of the initial inputs, the final step is to click the Generate Paths button to generate the path automatically. A scenario with varying intervals from the same ROI, start point, and finish point is given to validate this platform. This platform is evaluated using 1, 1.5, 2, and 2.5 m interval distance values. Figure 8 shows that the platform successfully auto-generating coordinates in four different scenarios. Furthermore, the coordinates generated by the path planning platform were validated by field tests to support the autonomous walk-behind Quick G1000 tractor.

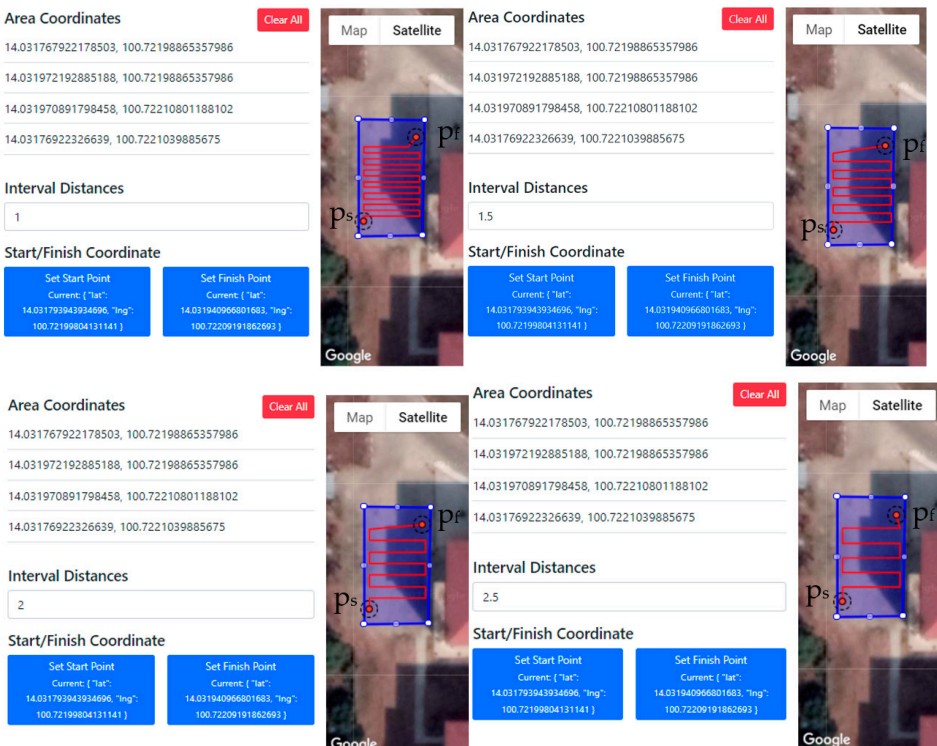

**Figure 8.** Path planning platform implementation using Laravel and Google Maps.

As a validation scenario, an Embedded System Platform was installed on the tractor. Figure 9 shows the Embedded System Platform box, which contains a controller and several motors that control the clutch handle and propel the tractor wheels autonomously via the waypoint method (coordinates from Path Planning Platform) at a constant and predetermined speed. In the controller, several algorithms are implanted to read heading distance, error, and angle so that the tractor can move autonomously, including Algorithms 2–4.

---

**Algorithm 2:** Compass Reading.

---

**Data:** normalize, normalize_YAxis, normalize_XAxis, heading, declinationAngle, headingDegrees
**Result:** headingDegrees
1.    normalize ← ReadCompassNormalize();
2.    heading = atan2(normalize_YAxis, normalize_XAxis);
3.    declinationAngle = ($\angle$ +(min/60))/(180/$\pi$);
4.    heading $-$ = declinationAngle;
5.       **if** (heading less than 0) **then**
6.          heading + = 2 × $\pi$;
7.       **else if** (heading more than 2 × $\pi$) **then**
8.          heading $-$ = 2 × $\pi$;
9.       **end if**
10.   headingDegrees ← ConvertToDegrees(heading);
11.   return headingDegrees;

---

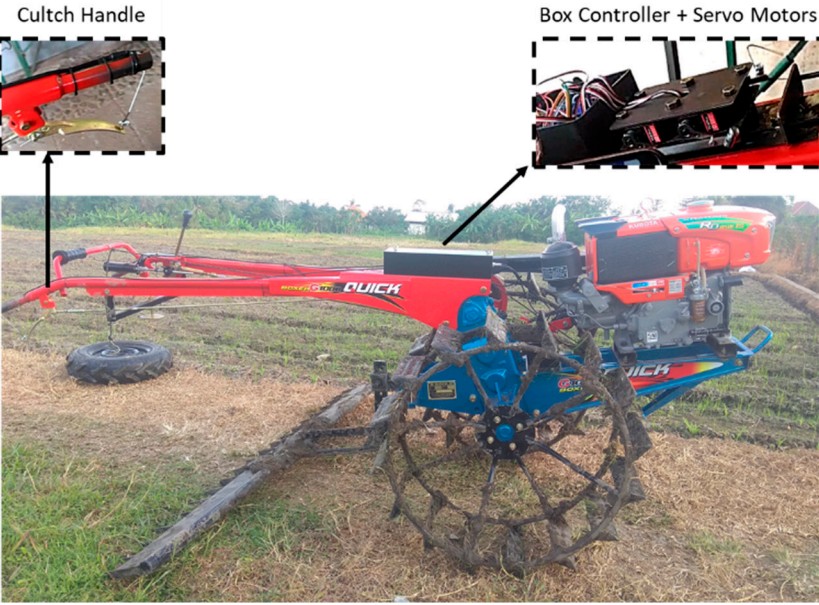

**Figure 9.** Quick G1000 walk-behind tractor.

Algorithm 2 is the compass sensor reading. The reading results are normalized to vector form and then used as input for the heading computation (atan2(normalize YAxis, normalize XAxis)). To compensate for the compass's divergence from the Earth's actual north pole, the declination angle ($\delta$), which may be found at http://magnetic-declination.com/ (1 February 2021), must be calculated. The declination angle for Gianyar, Bali, Indonesia is $0°43'$ EAST (POSITIVE). There are two conditions in which the heading calculation results should be corrected: larger than 2 × $\pi$ and fewer than 0 degrees. After that, the result is translated into degrees.

Algorithm 3 is a distance computation between the robot's current location and the target waypoint. The GPS sensor value represents the current location's degree of longitude and latitude, and the target waypoint must be translated to radians. Following that, the

Haversine formula is used to calculate the great-circle distance.

---

**Algorithm 3:** Calculate Distance to Target: distance from the current location to the target waypoint.

---

Data: deltaLongitude, currentLong, currentLat, targetLong, targetLat
Result: distanceToTarget
1.   $\Delta$Longitude $\leftarrow$ radians (currentLong $-$ targetLong);
2.   lat1 $\leftarrow$ radians(currentLat);
3.   lat2 $\leftarrow$ radians(targetLat);
4.   $\Delta$Longitude = sq ((cos(lat1) $\times$ sin(lat2)) $-$ (sin(lat1) $\times$ cos(lat2) $\times$ cos($\Delta$Longitude)));
5.   $\Delta$Longitude + = sq(cos(lat2) $\times$ cos($\Delta$Longitude));
6.   $\Delta$Longitude = sqrt(deltaLongitude);
7.   denom = (sin(lat1) $\times$ sin(lat2)) + (cos(lat1) $\times$ cos(lat2) $\times$ cos($\Delta$Longitude));
8.   $\Delta$Longitude = atan2($\Delta$Longitude, denom);
9.   distanceToTarget = $\Delta$Longitude $\times$ 6,372,795;
10.  return distanceToTarget;

---

---

**Algorithm 4:** Calculate robot turning to get to the waypoint target.

---

Data: headingError, targetHeading, currentHeading, errorOutput, headingTolerance, motorMovement
Result: motorMovement
1.   headingError = targetHeading $-$ currentHeading;
2.     **if** (headingError less than $-180$) **then**
3.        headingError + = 360;
4.       **else if** (headingError more than 180) **then**
5.        headingError $-$ = 360;
6.     **end if**
7.   errorOutput $\leftarrow$ calcErrorOutput(headingError)
8.   motorMovement $\leftarrow$ calcMotorMovement(errorOutput)
9.   return motorMovement;

---

Algorithm 4 is a heading error computation when the target heading value differs from the current heading value. The heading error calculation findings will be utilized as the input for the robot's movement, allowing it to operate autonomously. The direction of the tractor's heading is depicted in Figure 10. Two restrictions limit the results of the tractor heading angle calculation.

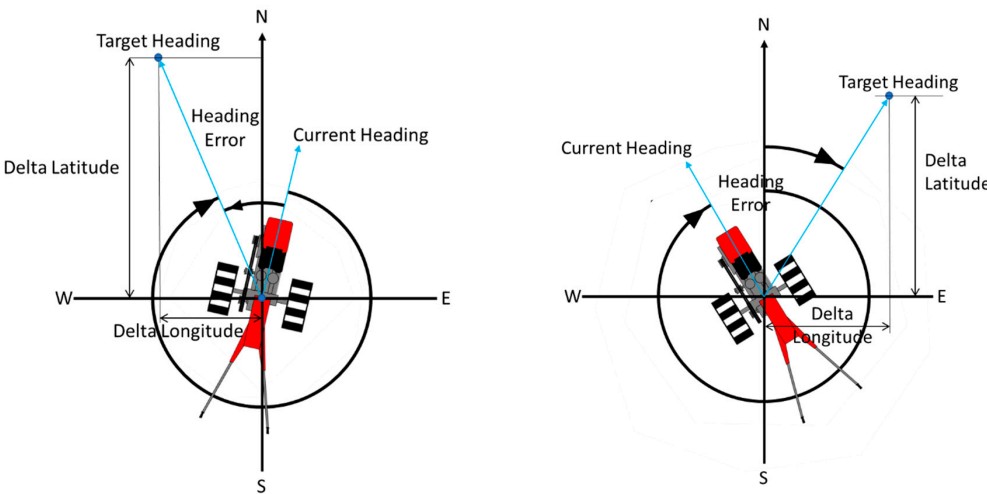

**Case 1:** Target Heading–Current Heading > 180          **Case 2:** Target Heading–Current Heading > $-180$

**Figure 10.** Tractor heading error.

Field tests were conducted and compared to simulation results to validate the Path Planning and Embedded System Platform. In this test, one scenario with the following characteristics was created:

1. Figure 11 depicts a waypoint's coordinates with ROI, start-finish points, and four-meter interval distance. The value of four meters is utilized since the accuracy of the employed GPS is insufficient to overcome interval values of less than four meters.
2. Field test results are compared with the tractor model simulation using the MatLab robotic toolbox with a differential drive kinematic model to determine the estimated plowing time.
3. The simulation parameters used to represent the G1000 Quick tractor are shown in Table 3.
4. The ROI and the waypoint coordinates used in the simulation are the same as the size generated from the Path Planning Platform.

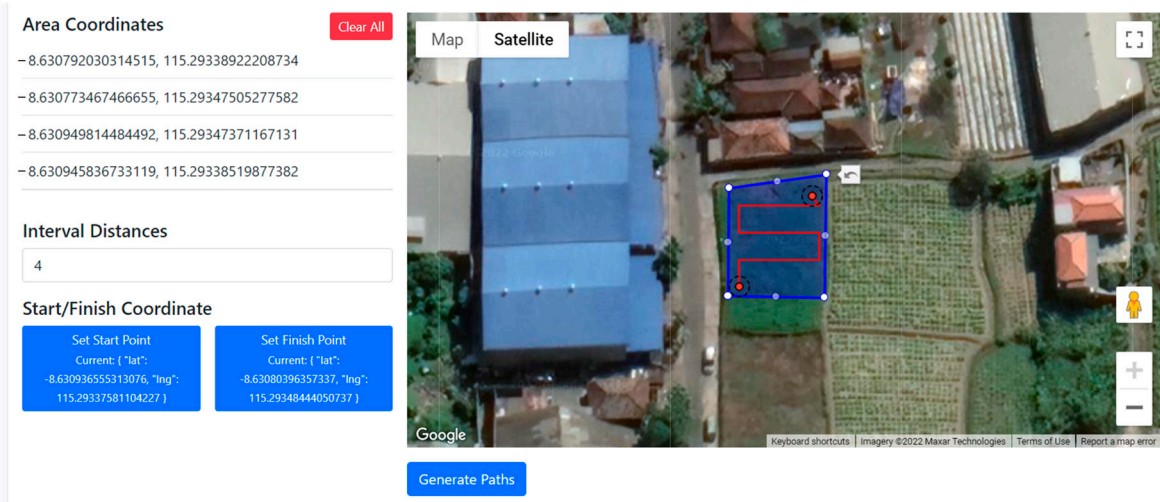

**Figure 11.** Field Trial Scenario with four-meter interval distance.

**Table 3.** Parameters for MatLab simulation.

| Parameter | Unit | Value |
|---|---|---|
| Linear Velocity | m/s | 0.56 |
| Wheel Radius | rad/s | $\pi/4$ |
| Distance between wheels | m | 1.8 |

Figure 12 shows the Simulink block diagram for the simulation. The waypoint block is an input containing an array of coordinates. The experimental scenario of the waypoint block has an array measuring $8 \times 2$. Based on the simulation results depicted in Figure 13, it is evident that the tractor follows the given coordinates of the path. This simulation shows that the estimated tractor movement time is 84 s. The simulation results are then compared with the tractor movement time in field trials, as shown in Table 3.

Field trials were carried out in one of the rice fields in the Ketewel area, Gianyar Regency, Bali, Indonesia. We used the same region of interest, starting point, finish point, and four meters distance interval. Each point is recorded using an IoT (Internet of Things) based logger module installed on the tractor. The data are stored in the database and processed using a text editor into gpx format, then visualized using the Google Earth application in Figure 14. The white line and red dots represent the robot's target path and waypoints, while the yellow dots represent the tractor path and the recording results of the logger module. This experiment demonstrates how a robot can calculate tractor orientation and pass waypoints. As seen in Figure 14, the highest error distance between

the target waypoint and the tractor position based on the GPS and Kalman Filter is 2.61 m. This is due to the type of GPS module (U-Blox Neo-M8N) used, which has an accuracy of 2.5 m. We also recorded the time data of plowing carried out in simulations and field tests with an area of interest of 302.65 m$^2$, and the results are shown in Table 4. The time required to generate the path planning coordinates is 685 milliseconds, while the time between simulation and trial is different. There is a difference of a few seconds in which the simulation conditions are faster than the actual field trial.

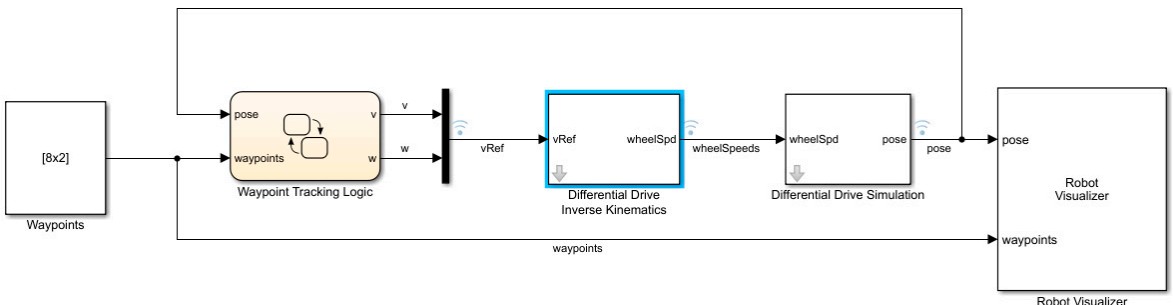

**Figure 12.** Matlab/Simulink waypoint simulation block diagram.

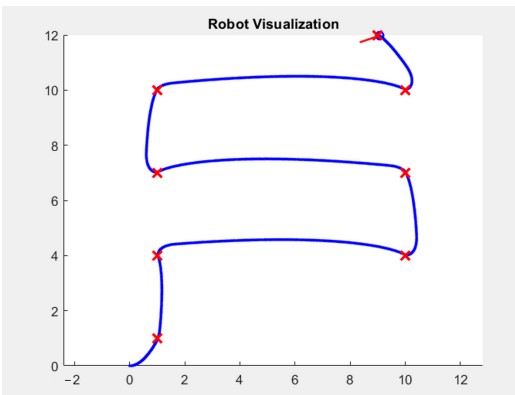

**Figure 13.** Robot movement simulation results. Visualization of robot movement using four meters distance between lines.

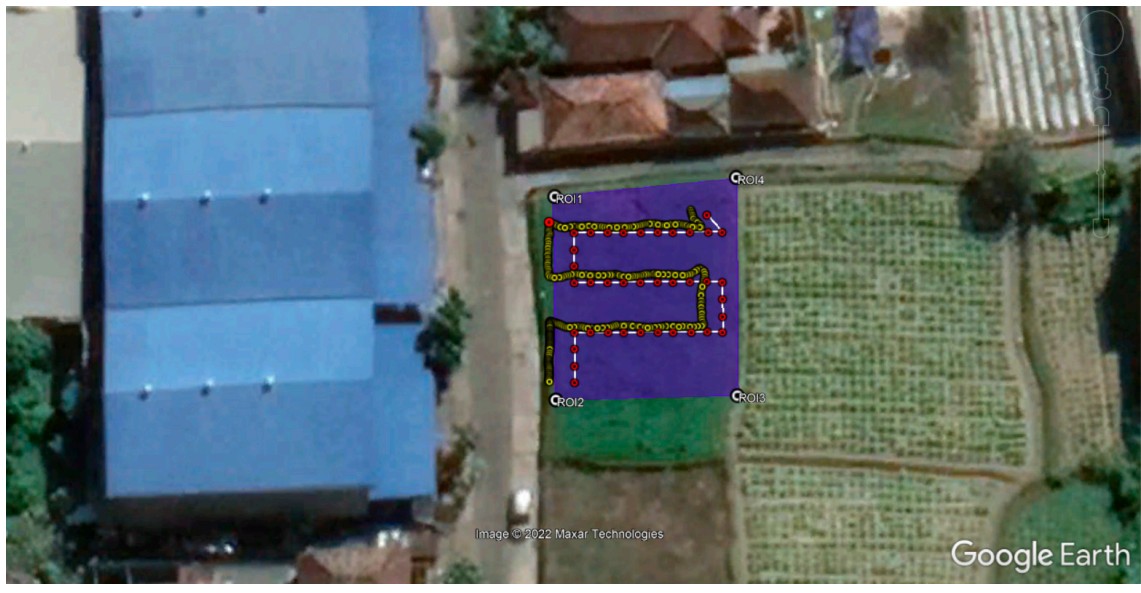

**Figure 14.** Data visualization on GPS navigation.

**Table 4.** Comparison of plowing process time between simulation results and field trials.

| Polygonal ROI | Processing Time | Simulation Tillage Time | Tillage Time |
|---|---|---|---|
| Test 1 | 685 ms | 84 s | 132 s |

## 5. Conclusions

A two-dimensional path planning platform using an edge-vertex path algorithm based on Laravel and Google Maps for the Autonomous Walk-Behind Hand Tractor has been presented. The algorithm automatically generates paths by considering the path interval distance (puddler width), start point, and finish point. Field trials were conducted with a walk-behind tractor to validate the resulting waypoint coordinates for rice field plowing missions. By applying the Kalman Filter, interference from GPS signals and magnetometer noise can be eliminated in certain instances. This system combines the Haversine formula and heading error calculation to automate waypoint navigation in tractors. The proposed algorithm permits farmers to specify the distance between paths based on varying puddler widths. Future research aims to design a path with a spiral pattern and to account for external disturbances, such as land contours, when constructing the path. In addition, the use of IoT technology is possible in the future because this path-planning platform can be implemented in the cloud.

**Author Contributions:** Conceptualization, P.N.C. and D.M.; methodology, P.N.C. and D.M.; software, P.N.C. and D.M.; validation, P.N.C. and D.M.; formal analysis, P.N.C. and D.M.; investigation, P.N.C. and D.M.; resources, P.N.C. and D.M.; data curation, P.N.C. and D.M.; writing—original draft preparation, P.N.C.; writing—review and editing, P.N.C. and D.M.; visualization, P.N.C.; supervision, D.M.; project administration, D.M. All authors have read and agreed to the published version of the manuscript.

**Funding:** This research received no external funding.

**Institutional Review Board Statement:** Not applicable.

**Informed Consent Statement:** Not applicable.

**Acknowledgments:** We would like to express our deepest gratitude to the Ministry of Research, Technology and Higher Education of the Republic of Indonesia, the STIKOM Bali Institute of Technology, and the Business and Rajamangala University of Technology Thanyaburi (RMUTT) for the support and facilities that were provided for this research.

**Conflicts of Interest:** The authors declare no conflict of interest.

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
