# Peer review of "Two-Dimensional Path Planning Platform for Autonomous Walk behind Hand Tractor"

_agriculture, doi:10.3390/agriculture12122051_

Round 1
Reviewer 1 Report
This main research aims to compute the coverage path equation (22) to obtain an edge-vertex path with a Boustrophedon Cellular Decomposition pattern and test it on the QUICK G-1000 walk-behind tractor. A path planning concept for back and forth path patterns is proposed, which con-siders the user's desired distance interval, start point, and finish point. The reviewer thinks that the algorithm proposed in this manuscript is practical and creative, but some words seem inappropriate. For example, 'Experience 1' in Table 4 should be 'Test 1'; in addition, why can the deviation shown in Figure 14 reach 2.61m? The accuracy is not high. I wonder why? Most importantly, the reviewer believed that the innovation of the manuscript should be listed in sections, and the refining depth of its innovation points was not enough, so it should be strengthened in conclusion section.
Reviewer 2 Report
The article is not scientific in nature. However, it addresses the very important issue of ensuring food sufficiency. Authors identified the need to implement autonomous machines, due to the lack of workers in agriculture, by using better path-plannig methods. For this purpose, they designed, tested and presented a suitable technical solution. It is a relatively simple solution that is not ground-breaking compared to the advanced machinery currently in use but dedicated to the small walk-behind machines. It is not clear what optimisation criterion did the authors use to select this path planning method? I also recommend updating the references, there is many positions older than 10 years.
Reviewer 3 Report
1. The paper's structure must be reviewed, togheter with the enumeration of both sections and subsections.
2. The paragraph "Related works" should be included into the introduction as a state of the art analysis. Said section should be improved considering a critical analysis of the existining literature related to the main reasearch topic. Authors must highlight the problem statement, the main scientific contribution of the study and the research objectives.
3. Figure 5 must be improved. For each subfigure, the following details must be added: caption, unit of measurements, and labels of each axis.
4. Conclusions must be improved, adding the results obtained with respect to the reasearch objective
Round 2
Reviewer 3 Report
In the revised version of the paper, the Authors took into account the observations of the previous review.